# Policies on mental health in the workplace during the COVID-19 pandemic: A scoping review

**David Villarreal-Zegarra** **\*, C. Mahony Reátegui-Rivera, Iselle Sabastizagal-Vela, Miguel Angel Burgos-Flores, Nieves Alejandra Cama-Ttito, Jaime Rosales-Rimache**

Centro Nacional de Salud Ocupacional y Protección del Ambiente para la Salud, Instituto Nacional de Salud, Lima, Peru

\* dvillarreal@ins.gob.pe, davidvillarreal@ipops.pe

## Abstract

### Background

The COVID-19 pandemic has had a profound impact on both mental health and working conditions. Workplaces are conducive spaces for implementing strategies and interventions to promote mental health. In addition to this, they are preventing, identifying, and managing mental disorders effectively. Although international agencies have identified some guidelines for the management of mental health in the workplace in the context of the COVID-19 pandemic, a more precise characterization of both the components of the policies, their implementation, and evidence of the outcome is required to provide useful information for decision-makers.

### Objectives

This study aims to synthesize scientific information regarding national and local policies focusing on preventing or improving, directly or indirectly, mental health problems in the workplace during COVID-19 pandemic.

### Methods

Our study is a scoping review. The Scopus, Web of Science, and Embase databases and PubMed search engine were used. Original and reviewed articles published from January 1, 2020 to October 14, 2021 were included in the research. Articles with abstract or full text in English, Spanish, German and Portuguese were also included. Our strategy is based on identifying policies (intervention) which focuses on directly or indirectly preventing or ameliorating mental health problems in the workplace during COVID-19 pandemic (participants).

### Results

A total of 6,522 records were identified, and only four studies were included in the scoping review, which were of low quality. That is, we found limited evidence evaluating mental health policies using primary or secondary data (empirical evaluation). Among the policies

**Data Availability Statement:** Our study is not a collection of primary or secondary data. We conducted a review of peer-reviewed scientific

articles that are available online. Although we do not attach the full-text documents, we do provide the title, authors, and details of the journal where it was published. Thus, it is feasible for interested researchers to access the articles included.

**Funding:** This research did not receive any specific grant from funding agencies in the public, commercial, or not-for-profit sectors. This study was conducted within the functions of the "Centro Nacional de Salud Ocupacional y Protección del Ambiente para la Salud" (National Center of Occupational Health and Environmental Protection for Health) of the "Instituto Nacional de Salud" (National Institute of Health) from Peru. The funders had no role in study design, data collection and analysis, decision to publish, or preparation of the manuscript.

**Competing interests:** The authors report no conflict of interest when conducting the study, analyzing the data, or writing the manuscript.

that have been identified are the increase of mental health resources, the promotion of mental health and self-care support programs, and the reduction of barriers to access to mental health treatment.

## Conclusion

Our research finds that there is limited evidence available to evaluate national and local policies aimed at directly or indirectly preventing or ameliorating mental health problems at work during COVID-19 pandemic. This forces decision-makers to use different criteria to guide the allocation of resources and budgets. Therefore, there is a need for health intelligence teams in health systems to be able to assess the impact of policies as an important input for decision-makers.

## Background

Mental health and work have a complex relationship. On one hand, working conditions and psychosocial risk factors can negatively affect mental health of workers, exacerbating or triggering health problems such as job stress, depression, anxiety, burnout, suicidal ideation, substance abuse, among other outcomes in mental health [1–3]. On the other hand, those individuals who suffer from a mental pathology may see their work dimension significantly affected, expressed as a decrease in productivity, absenteeism or presenteeism, and work injuries or accidents [4–6]. In addition, it has been hypothesized that mental health and working conditions are influenced reciprocally over time [7,8].

The COVID-19 pandemic has profoundly impacted both mental health and working conditions. Various studies have reported a high prevalence of mental health disorders in the general population that have been directly and indirectly attributed to COVID-19 pandemic [9–11]. Different systematic reviews have identified an increase in the prevalence of mental health problems in many countries for the general population and workers [12,13], which represents a problem at the public health level since a higher prevalence of mental health problems will overload the health system [14]. Likewise, a burden of mental health problems has a negative economic impact as it would reduce productivity, increase absenteeism and reduce the competitiveness of countries [15]. Therefore, it is necessary to develop strategies and policies to reduce the impact on population as a whole, especially on workers (productive companies of the country). The measures adopted by governments to mitigate the spread and transmission of the SARS-CoV-2 virus in the population, such as quarantines, blockades, social distancing, reduction of capacity, implementation of remote work, a restart of activities work in phases, among others, have drastically modified the working conditions of various sectors [16,17]. An example of this is the health sector, which, although it never stopped its activities because it was considered as an essential sector, was seriously affected by the pandemic since a substantial increase in the workload, maintained direct exposure to COVID-19, suffered a shortage of personal protective equipment, abuse by patients, stigma, and discrimination, amongst other problems [18–20].

The World Health Organization (WHO) notes that workplaces are important targets for the implementation of mental health prevention and promotion programs [21]. Likewise, the evidence supports this claim by reporting that workplaces are conducive spaces to implement strategies and interventions to promote mental health, in addition to preventing, identifying,

and managing mental disorders effectively [22,23]. However, there are still few policies that efficiently address mental health in the workplace [24], being even rarer in the context of the pandemic [25]. While international agencies have identified some guidelines for the management of mental health in the workplace in the context of COVID-19 pandemic [26], a more precise characterization is required, both of the components of the policies, their implementation, and the evidence of their results to have useful information for decision-makers. Therefore, the objective of this study is to synthesize scientific information regarding national and local policies focused on directly or indirectly preventing or improving mental health problems in the workplace during COVID-19 pandemic.

## Methods

### Design and protocol

Our study is a scoping review of scientific articles evaluating mental health policies in workers during the pandemic context. The PRISMA guidelines for Scoping Review (PRISMA-ScR) are followed, and compliance with all these criteria can be found in Supplementary Material 1 in S1 File.

The study protocol could not be registered in PROSPERO since this repository does not support scope review. Also, our protocol has not been registered in any repository.

### Search strategy

Our strategy is based on identifying policies (Intervention) which focuses on directly or indirectly preventing or improving mental health problems, in the workplace during the COVID-19 pandemic (participants). A search strategy was designed based on the identification of keywords related to policies, mental health (that is, mental health in general, anxiety, depression, and stress), COVID-19, and work context. In addition, terms related to these keywords available in the thesauri of scientific databases were used. The Scopus, Web of Science, and Embase databases were used, and the Pubmed search engine was used. The complete search strategies that were used are attached as supplementary material 2 in S1 File.

The search strategy for COVID-19 was based on recommendations made by the University of Medicine and Health Sciences (https://libguides.rcsi.ie/covid19/searchstrategy). On the other hand, the health policy search strategy was built on a scoping review carried out in China [27].

### Eligibility criteria

Original and review articles published during the period from January 1, 2020 to October 14, 2021 (date of the search) that addressed national and local policies promoted by official governments, whereby they are under execution or that had been executed, and that they were directly or indirectly related to mental health in workplaces. Articles with abstract or full text in English, Spanish, German and Portuguese were included.

### Study selection

All studies that met the following characteristics were included:

- Documents that made direct or indirect reference to workplace contexts during COVID-19 pandemic

- The intervention or exposure is an implemented national and local policy and the objective was to directly or indirectly improve mental health in the work context of any occupational

group. Studies on institutional policies in specific locations such as hospitals, schools or specific workplaces were excluded.

- The outcomes are any processed indicator, clinical outcome and the result of the impact of the policies, for instance, number of occupational psychology services deployed, additional amounts to health budgets, number of mental health care through tele consultations, the prevalence of mental health problems, beneficiaries' interviews, amongst others

- The papers presented primary or secondary data (quantitative, qualitative, or mixed) focusing on mental health policy outcomes. Also, narrative reviews, systematic reviews, and meta-analyses were also included, provided they focus on national or local mental health policies in the workplace.

Papers not related to the topic explored, duplicates, documents that presented only recommendations or theoretical aspects of policies (frameworks or reference frameworks), documents that only describe the policy formulation process, and protocols were excluded.

For this study, 6 reviewers were trained to homogenize the criteria for selecting articles and extracting information. Before the beginning of the review by title and abstract, previous training was carried out to calibrate the selection of documents by selecting a subsample of 20 articles that all the reviewers evaluated. Disagreements were resolved among the entire group. After this training, all title and abstract documents were divided among the reviewers. At least two reviewers independently reviewed each document for the title and abstract and full text. Conflicts were resolved in the first instance by the reviewers; and in the second, by a third reviewer who decided whether the document was included or excluded.

## Data items

The following data were collected:

- The surname of the first author.

- Country where the policy was enacted.

- The implementation date of the policy.

- Policy objective (i.e., surveillance, prevention of mental disorders, promotion of mental health, treatment of mental health problems, and others).

- A summary of the policy.

- Target population (i.e., health personnel, military, general population, etc.).

## Synthesis of results

A narrative description of the included studies was made. The results were grouped whenever possible, according to the information on the implemented policies and their implementation process.

## Risk of bias

Different tools were used to assess the risk of bias, as high heterogeneity of the included study designs is expected. We used the Joanna Briggs Institute (JBI) checklist bias tools: Checklist for Analytical Cross-Sectional Studies, Checklist for Case Control Studies, Checklist for Cohort Studies, Checklist for Qualitative Research, Checklist for Quasi-Experimental Studies,

Checklist for Randomized Controlled Trials, and Checklist for Systematic Reviews (https://jbi.global/critical-appraisal-tools). Narrative studies did not have a risk of bias assessment.

## Ethics statement

The study is a secondary literature review, so no primary data collection was performed and it does not involve ethical risk. Our study protocol was not submitted to an ethics committee because, since we did not collect primary data from the participants, it did not put any person at risk. It should be noted that informed consent was not required, since no primary data was collected, because our study only uses results already published in scientific articles.

## Results

### Selection of study

A total of 6,522 documents were identified, after filtering for duplicates the number dropped to 3,129 unique documents. A total of 3,082 were removed in the title and abstract review, and 43 were removed in the full-text review. Finally, four study was included in the scoping review (see Fig 1). The reasons for the exclusion of each of the studies that passed full text are attached in supplementary material 3 in S1 File.

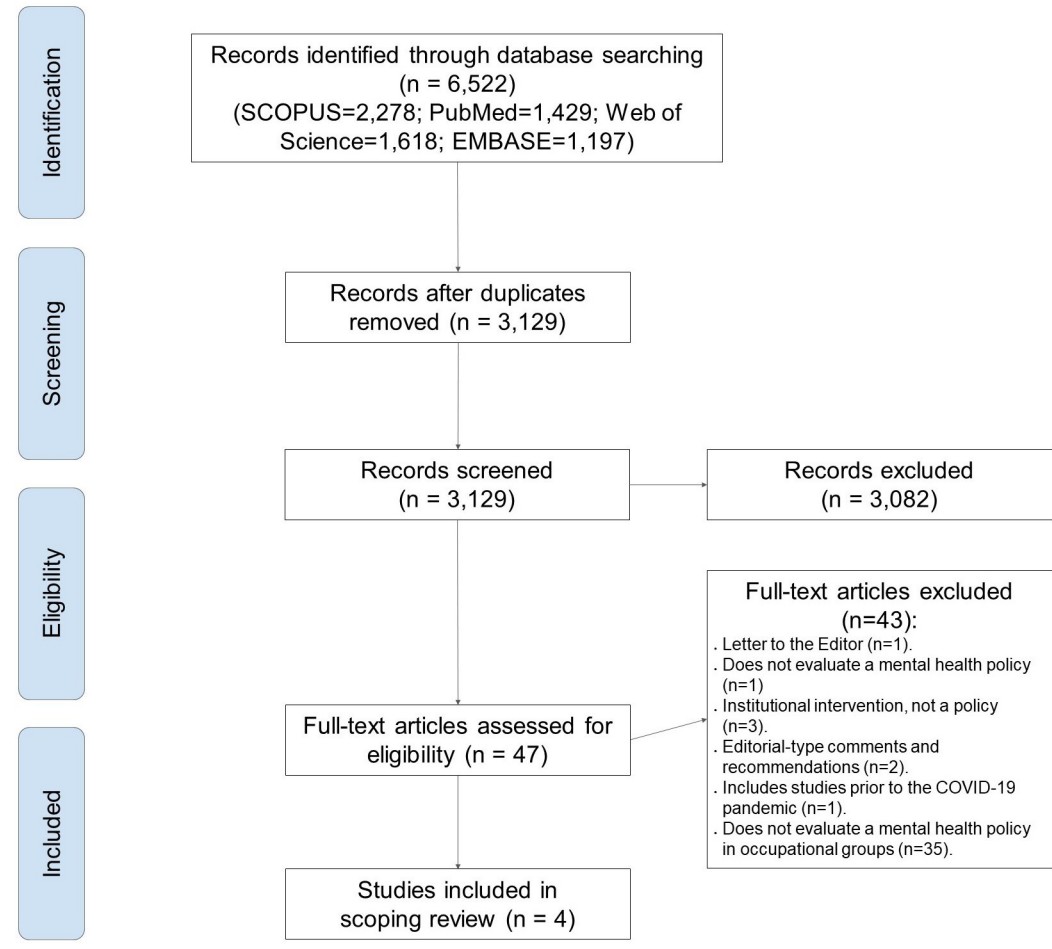

**Fig 1. Flowchart according to PRISMA.**

## Characteristics of the included studies

Two of the included studies are narrative, one is cross-sectional, and one is a systematic review. In addition, three of the studies were conducted in China and one in the United States. Table 1 details individually the objectives of the policies evaluated in each study, a summary of the study, and the target audience for the policy.

## Risk of bias

We only performed a risk of bias analysis for two of the five included studies. We identified the risk of bias for the cross-sectional study [28] and the systematic review [29] as high in both cases (see Table 2). We found two main sources of bias in the cross-sectional study. First, the setting and subjects who benefited from the policy were not clearly described. Second, although some confounders were controlled for using structural equation modeling, there are several other sources of confounding that could not be controlled for and were not accounted for. On the other hand, in the systematic review, the databases where the document search was performed were limited, and it is not clear the extraction methods and whether the document review was performed in duplicate. It should be noted that risk of bias analysis was not performed on the narrative studies [30,31].

**Table 1. Studies included in the scoping review (n = 4).**

| Last name of the first author | Country where the policy was enacted | Design (Date of implementation) | Policy objective | Policy Summary | Target population |
|---|---|---|---|---|---|
| Wong [28] | China | Cross-Sectional (February 2020) | To explore the relationship between employee's view on workplace policy, perceived likelihood of risk and Health-Related Quality of Life in working population during COVID-19 pandemic. | The study analyzes employee's view on workplace policies to protect their health in terms of comprehensiveness, timeliness and transparency.Workplace Policy:<br>• Workplace infection control and prevention policy.<br>• Workplace measures by the supply of protective equipment.<br> Of 1048 respondents, 16% reported that no workplace measures nor guidelines were existed in their company related to the COVID-19 pandemics. Those who reported having workplace policy were not satisfied with the arrangement in term of comprehensiveness (36%), timeliness (38%), and transparency (63%). Regarding to the policy measure, only 68% respondents reported that their workplace supplied face masks to them. The health index was 0897, which was lower than the norm of 0.924. 64% of respondents reported having a health problem in at least 1 of 5 dimension of EQ-5D-5L with the highest proportion of having problem in anxiety/depression (55%). In addition, the workplace policy and measure had a direct effect of 0.131 on health outcome. Perception of infection risk had a direct effect of 0.218 on health outcome and partly mediated the relationship between workplace policy and measure and health outcome (0.066).The study highlighted the negative impact on health-related quality of life associated with the lack of workplace policies, lack of protective equipment provision, and dissatisfaction with workplace policies. | Hong Kong employees |
| Zhang [29] | China | Systematic review (January 2020 to May 2020) | To (1) describe the psychological status of medical workers at different time points during the COVID-19 pandemic in China and (2) to preliminarily explore the impact of national policies on the psychological well-being of medical workers. | A series of related national policies have been issued to promote mental wellness care among healthcare workers.<br>The National Disease Control and Prevention Bureau launched the "Principles for Emergency Psychological Crisis Intervention for the COVID-19 Pandemic", which emphasized the priority of psychological support and intervention for frontline medical staff. The National Health Commission launched a policy requiring psychiatric medical personnel to support Hubei Province, and specific medical service lines and special areas were set up in the existing psychological hotline and psychological assistance network.<br>Several policies were issued to improve the care of medical personnel in terms of security, work environment, family needs, and psychological support. There were also others aimed at further improving care for medical workers and their families. Although no studies quantify the impact of each policy or measure, their positive effect is evident from the changes in the psychological status of medical workers. | Chinese healthcare workers (medical staff, such as doctors, nurses and technicians) |

*(Continued)*

**Table 1.** (Continued)

| Last name of the first author | Country where the policy was enacted | Design (Date of implementation) | Policy objective | Policy Summary | Target population |
|---|---|---|---|---|---|
| Goldman [30] | USA | Narrative study (April 2020) | Promote and enable changes in the provision of mental health care in the face of the COVID-19 crisis. It is done through legislation, regulation, financing, accountability, and workforce development. | Legislation: The Coronavirus Relief, Relief and Economic Security Act (CARES) includes $ 425 million appropriations for the Substance Abuse and Mental Health Services Administration (SAMHSA) to respond to the pandemic, with $ 250 million earmarked to new funding for Community Certified Behavioral Health (CCBHC) expansion grants, $ 100 million for emergency response activities, and $ 50 million for suicide prevention (Division B, Title VIII).<br>• Regulation: A wide range of regulations have been issued in response to the COVID-19 crisis, most of which are aimed at reducing the requirements for face-to-face contact between patients and providers to minimize viral transmission.<br>• Funding: The Centers for Medicare & Medicaid Services (CMS) issued a "blanket waiver" 1135 to allow greater flexibility in Medicare and Medicaid reimbursements, reduce pre-authorizations, and allow easier transfer of patients between facilities, all of which will support the provision of mental health care during the pandemic.<br>• Accountability: The Centers for Medicare & Medicaid Services (CMS) have delayed quality reporting requirements for programs that require quality reporting, such as the Merit-Based Incentive Payment System, which includes healthcare providers mental.<br>• - Workforce development: Measures to strengthen the workforce in the face of COVID-19 have primarily focused on maximizing access to mental health providers while reducing the administrative burden. | General population. It does not specify the working population (except for first-line health personnel, it is indicated that since April 2020 they can access paid sick leave if they have symptoms of COVID-19, need to be in quarantine, or are caring for sick children or relatives, this according to the First Family Coronavirus Response Act (HR 6201). |
| Ju [31] | China | Narrative study (January 2020) | Address the widespread mental health needs arising from this pandemic. | The response of the system was divided into groups differentiated teams:<br>• The team of mental health experts is made up of mental health experts in crisis intervention and has three main responsibilities: providing policy-making consultations to the national or provincial Joint Prevention and Control Mechanism, offering professional training and supervision for social volunteers, and provide psychoeducation. For the general public.<br>• The psychological rescue team comprises psychiatry and psychology personnel, as well as liaison officers. This team is primarily responsible for providing health education, intervention in psychological crises, consultation psychiatry service, and liaison both for healthcare workers as well as for patients inside the wards.<br>• The counseling team includes psychologists and professionals with experience in psychological crisis intervention and they take on the role of online counseling.<br>• Social support forces are made up of social workers, non-governmental organizations, and social volunteers, who provide psychosocial support during the epidemic.<br> The strategies:<br>1. Dissemination of information: reliable from the state, accurate and updated data on the state of the pandemic, mental health education, through the media and official websites.<br>2. Public education: online education on mental health, according to population subgroups and with an updated bibliography.<br>3. Evaluation and intervention: online self-assessment of mental well-being, with free online support for those who obtained values above the cut-off point, there were mental health and counseling hotlines.<br>4. Mental health and training services for healthcare workers: healthcare workers preparing for front-line clinical work receive a short mental health training to equip them with the basic knowledge and skills to identify and refer hospitalized patients in need of more mental health care. The training also provided some self-help techniques to improve your mental health resilience in the face of psychological stress associated with caring for patients with serious infections. In addition, mental health professionals were working at full capacity to support frontline hospital workers. Notifications were sent out regularly to inform you of available mental health care,<br>5. Mental Health Services for Patients: Newly admitted patients receive brochures that educate them on common mental health problems associated with COVID-19 and ways to address these mental health needs. Additionally, various types of group therapy activities have been integrated into daily ward routines to alleviate loneliness, boredom, and frustration caused by infection, as well as prolonged quarantine periods. Due to the strict isolation measures in these isolation rooms, mental health professionals provide psychological counseling to infected patients primarily through online and telephone means. For patients with serious mental health risks, such as suicide attempts or major behavioral disorders, | Four different populations with different levels of mental health needs according to the intensity of psychological stressors related to the COVID-19 epidemic: (A) COVID-19 hospitalized patients, front-line health personnel, and other personnel who have supported by the first line of prevention and control of the epidemic. (B) Patients in quarantine due to a confirmed or suspected diagnosis of COVID-19 with mild symptoms and patients with fever. (C) Family members or friends of the first two risk levels and the rest of the personnel for the control and prevention of the epidemic. (D) People who have been affected by pandemic prevention and control measures by areas, susceptible population, and the general public. |

**Table 2. Checklist of Joanna Briggs Institute (cross-sectional and systematic review).**

| Cross-Sectional Studies | Wong [28] |
|---|---|
| 1. Were the criteria for inclusion in the sample clearly defined? | + |
| 2. Were the study subjects and the setting described in detail? | - |
| 3. Was the exposure measured in a valid and reliable way? | - |
| 4. Were objective, standard criteria used for measurement of the condition? | + |
| 5. Were confounding factors identified? | - |
| 6. Were strategies to deal with confounding factors stated? | - |
| 7. Were the outcomes measured in a valid and reliable way? | + |
| 8. Was appropriate statistical analysis used? | + |
| **Systematic review** | Zhang [29] |
| 1. Is the review question clearly and explicitly stated? | + |
| 2. Were the inclusion criteria appropriate for the review question? | + |
| 3. Was the search strategy appropriate? | + |
| 4. Were the sources and resources used to search for studies adequate? | - |
| 5. Were the criteria for appraising studies appropriate? | ? |
| 6. Was critical appraisal conducted by two or more reviewers independently? | ? |
| 7. Were there methods to minimize errors in data extraction? | ? |
| 8. Were the methods used to combine studies appropriate? | NA |
| 9. Was the likelihood of publication bias assessed? | NA |
| 10. Were recommendations for policy and/or practice supported by the reported data? | + |
| 11. Were the specific directives for new research appropriate? | + |

Note: + = Yes.— = No. "?" = Unclear. NA = Not applicable.

## Synthesis of results

**Increase in health resources.** A study indicates that different policies of economic vouchers or service subsidies have been developed that aim to financially compensate the losses produced by the context of the pandemic and facilitate access to basic services such as health services [30]. That is why the US government has allocated about $ 2 trillion, with multiple provisions, to mental health providers [30], including $ 425 million in appropriations for the Substance Abuse and Mental Health Services Administration to respond to the pandemic, $ 250 million earmarked for new funding for Certified Behavioral Health Community Clinic expansion grants, and $ 50 million for suicide prevention activities [30].

**Reduction of barriers to care.** The US government implemented different strategies to reduce barriers to mental health care for the general population and workers. First, telehealth regulations were temporarily relaxed, allowing payments and exempting copayments for services provided to beneficiaries in all clinical areas, regardless of whether there was a relationship with the health service provider previously [30]. Second, the temporary exemption allowed providers to provide telehealth using technology platforms that did not comply with the Health Insurance Portability and Accountability Act to reduce barriers to care [30]. Third, measures to strengthen the workforce in the face of COVID-19 have primarily focused on maximizing access to mental health providers while reducing the administrative burden. With the coordination of the Federation of State Medical Boards, many states have temporarily waived state licensing and renewal requirements and allowed for greater reciprocity in the US [30].

On the other hand, a cross-sectional study has highlighted the perception of workers on the implemented policy. So it is not only the need to implement these policies, but a positive

perception of the transparency and relevance of policies can have a positive impact on people's quality of life [28].

**Mental health support and self-care.** In China, health workers preparing for front-line clinical work in mental health were trained to equip them with the basic knowledge and skills to identify and refer hospitalized patients and other mental health professionals in need of mental health care [31]. In addition, health personnel was trained in self-help techniques to improve the resilience of their mental health in the face of the psychological stress associated with the care of patients with serious infections [31].

A meta-analysis suggested that national policies of targeted psychological support for health care workers and frontline health care workers may have an impact on the prevalence of depressive symptoms, anxiety, stress, and sleep problems during the context of the COVID-19 pandemic [29]. In addition, a narrative reviews have also reported that policies promoting and regulating mental health interventions in health professionals have had a positive impact within the context of the pandemic [31].

## Discussion

During the context of the COVID-19 pandemic, different governments worldwide have deployed a variety of policies to reduce infections and deaths from the virus within the workplace [32]. However, the number of policies that have directly or indirectly aimed to improve workers' mental health or address mental health needs in the workplace during the pandemic is still limited [25,30,31,33], and their actual impact on workers has yet to be assessed. Policies that have been implemented to indirectly improve mental health include, for example, increasing the number of resources for the mental health sector; reducing barriers to accessing mental health care during the pandemic context; and training lay staff in basic mental health care and self-care tools [30,31]. However, we were unable to locate any high-quality studies with primary or secondary data that have evaluated the effect of public policies focused on preventing or ameliorating, directly or indirectly, mental health problems in the workplace during the COVID-19 pandemic. The only study with primary data was of low quality and indirectly evaluates the effect of an infection control and prevention policy in China [28], but its exposure is employee opinion on workplace policies and the likelihood of perceived occupational hazards, and the outcome is health-related quality of life in the working population during the COVID-19 pandemic. On the other hand, a systematic review that evaluate the effect of an emotional support policy on health care workers, but using a cascade meta-analysis to evaluate cross-sectional measurements [29]. However, a systematic review is not the most appropriate design for evaluating the effect of a policy as such, as this would require a clinical trial.

Our scoping review found that there is limited evidence on the effect of workplace policies to improve mental health in workers, thus representing an opportunity to stimulate policy evaluation research related to mental health care within work contexts, especially those who are at a high level of exposure such as health professionals or police officers [34]. Some of the occupational elements that have an impact on workers' mental health in the context of COVID-19 are job insecurity, long periods of isolation and uncertainty about the future to have a direct impact on workers' mental health [25]. Therefore, organizational and work interventions can mitigate this scenario, such as improving workplace infrastructures, adopting anti-contagion measures such as the regular provision of personal protective equipment, postponing measures to return to face-to-face work, and implementing resilience training programs [25,34,35]. Although there is evidence that these isolated interventions can have an impact on the mental health of workers, it is not known whether when established as public policies they have an impact on the mental health of the population. Therefore, there is a need

for health intelligence teams in different countries to assess the impact of these public policies on workers' mental health.

Our study focused on mental health policies, however, other types of economic or health policies may have an indirect impact on the mental health of workers, without their central objective. For example, infection control policies [28]. Some specific policies that were found have being to keep the duration of quarantine as short as possible, to reinforce communication about the status of the pandemic, to provide official and adequate information about the pandemic context, and to provide supplies to cover the basic needs of those most in need [31,36]. These national and local policies have suggested having a positive impact on mental health in the working population, so it is necessary to consider them as part of the mental health response of health systems. In addition, the COVID-19 policy tracker designed by the University of Oxford provides a global view of the policies used in different countries to deal with the pandemic [32]. Although there is no mental health component as such in the tracker, it does positively value the presence of vaccination programs, policies for the use of personal protective equipment, or mass information campaigns; which can be indirect indicators of a response to mental health of population.

Currently, there are different frameworks (frames of reference) that seek to explain the response of health systems to the COVID-19 pandemic, in particular, two frameworks consider that the increase in resources in mental health and the reduction of barriers to access to mental health as necessary actions to improve mental health in the context of the pandemic [37,38]. On one hand, one of the frameworks designed from the experience of early career psychiatrists presents a framework for action and preparation of health systems to face the impact on mental health that the pandemic has [37]. This framework proposes a response and preparation based on five types of components: a) planning and coordination; b) monitoring and evaluation; c) reduction of mental health problems and misinformation; d) maintenance of mental health services; e) constant communication of this entire process [37]. Although it could be useful to develop health policies during the pandemic, it has the limitation that it only considers the first wave of COVID-19 [37]. On the other hand, another framework focused on the early impacts of the pandemic on mental health care and people with mental health problems, this one identified a series of necessary areas to consider when designing policies and other types of research [38]. Within these areas are the experiences of people with mental health problems (loneliness and isolation; lack of access to essential services and resources; family and social adversities; risk of COVID-19 infection; positive life experiences during the pandemic), strategies used by people with mental health problems to cope with the pandemic (self-care strategies; peer and community support), the impact of the health service (changes in service activity), challenges and adaptations of the service (challenges in hospital, residential and community settings; adaptations and innovations of the service), and the ethical challenges that involve [38]. Although both frameworks are useful for designing policies and other interventions, it has not been possible to identify literature that details the process of articulation of these policies within the health system, so decision-makers must consider the complexity of their health systems when it's the most appropriate time to implement mental health policies in an occupational context.

From the initial stages of the pandemic, different initiatives have been reported by private institutions, scientific societies, universities, and civil society to provide mental health assistance to the general population and technical support to governments [39–41]. However, it has not been identified that these initiatives have managed to escalate to the public policy. Although in some countries they have sought to map and articulate individual initiatives [39]. It is necessary to provide policies that articulate and direct initiatives within the health system, to provide comprehensive care and avoid duplication of efforts.

## Strengths and limitations

Our research achieves a comprehensive review of the available evidence on mental health policies in occupational settings during the COVID-19 pandemic. However, we have identified two main limitations. First, our review found limited available evidence (only two low-quality studies and two narrative studies); it is not possible to identify policies that have the greatest impact on workers' mental health. However, this may be because it is still too early for studies to have been published on the topic, so further review on the subject is needed. The research team invites different health intelligence teams to replicate the search in two to three years' time, when more evidence is available, considering that during the pandemic context, the evidence becomes outdated very quickly. Second, our search was conducted to identify scientific publications on the evaluation of government policies and standards for workers' mental health. Although we searched different databases, it was not within our objective to consider grey literature such as policies, standards, technical notes, or regulations from different countries. This decision may represent a partial review of available evidence, as policy evaluations could have been performed through technical standards or internal reports; however, these documents are not peer-reviewed and are not fully reliable. Because our objective is not to evaluate policies directly, but to evaluate the available evidence on national and local policies. Third, it is possible that, at the time of publication, the findings presented may partially represent all available scientific evidence due to the speed with which new studies are published within the context of COVID-19. Thus, it warrants ongoing updates of the review by health intelligence teams throughout the pandemic.

## Conclusions

Our research finds that there is limited evidence available to evaluate national and local policies aimed at directly or indirectly preventing or ameliorating mental health problems at work during COVID-19 pandemic. Among the policies that have been identified are the increase of mental health resources, the promotion of mental health and self-care support programs, and the reduction of barriers to access to mental health treatment. However, the evidence evaluating these policies is of low quality. This forces decision-makers to use different criteria to guide the allocation of resources and budgets. Therefore, there is a need for health intelligence teams in health systems to be able to assess the impact of policies as an important input for decision-makers.

## Supporting information

**S1 File.**
(DOCX)

## Author Contributions

**Conceptualization:** David Villarreal-Zegarra, C. Mahony Reátegui-Rivera, Iselle Sabastizagal-Vela, Miguel Angel Burgos-Flores, Nieves Alejandra Cama-Ttito, Jaime Rosales-Rimache.

**Data curation:** David Villarreal-Zegarra.

**Formal analysis:** David Villarreal-Zegarra.

**Investigation:** David Villarreal-Zegarra, C. Mahony Reátegui-Rivera, Iselle Sabastizagal-Vela, Miguel Angel Burgos-Flores, Nieves Alejandra Cama-Ttito, Jaime Rosales-Rimache.

**Methodology:** David Villarreal-Zegarra, C. Mahony Reátegui-Rivera, Iselle Sabastizagal-Vela, Nieves Alejandra Cama-Ttito, Jaime Rosales-Rimache.

**Project administration:** David Villarreal-Zegarra.

**Supervision:** C. Mahony Reátegui-Rivera, Iselle Sabastizagal-Vela, Miguel Angel Burgos-Flores, Nieves Alejandra Cama-Ttito, Jaime Rosales-Rimache.

**Validation:** David Villarreal-Zegarra, C. Mahony Reátegui-Rivera, Iselle Sabastizagal-Vela, Miguel Angel Burgos-Flores, Nieves Alejandra Cama-Ttito, Jaime Rosales-Rimache.

**Visualization:** C. Mahony Reátegui-Rivera, Iselle Sabastizagal-Vela, Miguel Angel Burgos-Flores, Nieves Alejandra Cama-Ttito, Jaime Rosales-Rimache.

**Writing – original draft:** David Villarreal-Zegarra.

**Writing – review & editing:** C. Mahony Reátegui-Rivera, Iselle Sabastizagal-Vela, Miguel Angel Burgos-Flores, Nieves Alejandra Cama-Ttito, Jaime Rosales-Rimache.

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
