## [Decision Letter · Decision Letter 0]

30 Mar 2022

PONE-D-21-38416Mental health policies in the workplace during the COVID-19 pandemic: A scoping reviewPLOS ONE

Dear Dr. Villarreal-Zegarra,

Thank you for submitting your manuscript to PLOS ONE. After careful consideration, we feel that it has merit but does not fully meet PLOS ONE’s publication criteria as it currently stands. Therefore, we invite you to submit a revised version of the manuscript that addresses the points raised during the review process. Your paper has been scientifically judged by two referees. Overall, their comments are positive (to the extent that they highlight the potential value of this work for further interventions on workers' mental health), but several revisions are needed. You will find a (not very extensive) set of comments appended below, as well as an attached file (i.e., sanitized copy) of the paper, as commented by our Reviewer # 1. These comments are mostly related to theoretical issues, supporting sources and interpretations, and become relevant for this type of paper. Therefore, please consider carefully addressing all of them in the paper, also providing suitable responses and rationales in your rebuttal letter.

We look forward to receiving your revised manuscript.

Kind regards,

Sergio A. Useche, Ph.D.

Academic Editor

PLOS ONE

Journal Requirements:

This research did not receive any specific grant from funding agencies in the public, commercial, or not-for-profit sectors. This study was conducted within the functions of the "Centro Nacional de Salud Ocupacional y Protección del Ambiente para la Salud" (National Center of Occupational Health and Environmental Protection for Health) of the "Instituto Nacional de Salud" (National Institute of Health) from Peru.

This research did not receive any specific grant from funding agencies in the public, commercial, or not-for-profit sectors. This study was conducted within the functions of the "Centro Nacional de Salud Ocupacional y Protección del Ambiente para la Salud" (National Center of Occupational Health and Environmental Protection for Health) of the "Instituto Nacional de Salud" (National Institute of Health) from Peru.

This research did not receive any specific grant from funding agencies in the public, commercial, or not-for-profit sectors. This study was conducted within the functions of the "Centro Nacional de Salud Ocupacional y Protección del Ambiente para la Salud" (National Center of Occupational Health and Environmental Protection for Health) of the "Instituto Nacional de Salud" (National Institute of Health) from Peru.

6. We note that this manuscript is a systematic review or meta-analysis; our author guidelines therefore require that you use PRISMA guidance to help improve reporting quality of this type of study. Please upload copies of the completed PRISMA checklist as Supporting Information with a file name “PRISMA checklist”.

Reviewers' comments:

Reviewer's Responses to Questions

**Comments to the Author**

1. Is the manuscript technically sound, and do the data support the conclusions?

Reviewer #1: Partly

Reviewer #2: Partly

2. Has the statistical analysis been performed appropriately and rigorously? 

Reviewer #1: N/A

Reviewer #2: No

3. Have the authors made all data underlying the findings in their manuscript fully available?

Reviewer #1: Yes

Reviewer #2: Yes

4. Is the manuscript presented in an intelligible fashion and written in standard English?

Reviewer #1: Yes

Reviewer #2: Yes

5. Review Comments to the Author

Reviewer #1: Thanks for the opportunity to review this manuscript. The authors have undertaken a scoping review to identify that impact of nationally implemented COVID-19 mental health policies within the workplace. Please see the attached document for a few points for consideration to strengthen your review.

Reviewer #2: The article performs a systematic review on the mental health policies implemented in the workplace during the COVID-19 pandemic. Clearly, this is an important topic, useful to know to what extent companies have involved themselves in the state of their workers, as well as to examine the efficacy (or lack of it) of the employed measures.

There is plenty of international evidence pointing out the mental health issues that have been increasing during the past few years (e.g. doi: 10.1186/s12992-020-00589-w). Therefore, the background must be complemented with data and references specifying the most relevant issues, clarifying the importance of the topic and its practical implications.

The systematic review cannot have one article only. It is true that, to the day, there are few papers analyzing the topic of policies implemented by companies. Therefore, I recommend slightly changing the eligibility criteria of the articles, in order to include at least 5 or 6 more of them in the itemization and analyses of contents. For instance, you could include studies analyzing the mental health issues derived from COVID-19 in specific populations of workers, even though they may not address the measures employed by the companies (e.g. doi: 10.7717/peerj.13050).

Also, a new search must be performed in the database, since this topic is current, and articles are being constantly published. Therefore, it is probable that since October 14th some articles have been published hat adjust to the criteria established by the authors, and they will have to be included in the review.

6. PLOS authors have the option to publish the peer review history of their article (what does this mean?). If published, this will include your full peer review and any attached files.

Reviewer #1: No

Reviewer #2: No

---

## [Author Response · Author response to Decision Letter 0]

1 Jun 2022

May 2022

Dear Editor,

We would like to thank you and the reviewers for your comments. We have carefully reviewed them and have revised the manuscript accordingly. Our responses are detailed below. Highlighted in red are the changes to the manuscript. We hope the revised version is now suitable for publication and look forward to hearing from you in due course.

Sincerely,

The authors,

-------

Journal Requirements:

Reply: We have confirmed that our manuscript meets the style requirements of PLOS ONE.

Reply: The Funding Information was resubmitted.

“This research did not receive any specific grant from funding agencies in the public, commercial, or not-for-profit sectors. This study was conducted within the functions of the "Centro Nacional de Salud Ocupacional y Protección del Ambiente para la Salud" (National Center of Occupational Health and Environmental Protection for Health) of the "Instituto Nacional de Salud" (National Institute of Health) from Peru. The funders had no role in study design, data collection and analysis, decision to publish, or preparation of the manuscript.”

This research did not receive any specific grant from funding agencies in the public, commercial, or not-for-profit sectors. This study was conducted within the functions of the "Centro Nacional de Salud Ocupacional y Protección del Ambiente para la Salud" (National Center of Occupational Health and Environmental Protection for Health) of the "Instituto Nacional de Salud" (National Institute of Health) from Peru.

Reply: The section was modified:

“This research did not receive any specific grant from funding agencies in the public, commercial, or not-for-profit sectors. This study was conducted within the functions of the "Centro Nacional de Salud Ocupacional y Protección del Ambiente para la Salud" (National Center of Occupational Health and Environmental Protection for Health) of the "Instituto Nacional de Salud" (National Institute of Health) from Peru. The funders had no role in study design, data collection and analysis, decision to publish, or preparation of the manuscript.”

This research did not receive any specific grant from funding agencies in the public, commercial, or not-for-profit sectors. This study was conducted within the functions of the "Centro Nacional de Salud Ocupacional y Protección del Ambiente para la Salud" (National Center of Occupational Health and Environmental Protection for Health) of the "Instituto Nacional de Salud" (National Institute of Health) from Peru.

This research did not receive any specific grant from funding agencies in the public, commercial, or not-for-profit sectors. This study was conducted within the functions of the "Centro Nacional de Salud Ocupacional y Protección del Ambiente para la Salud" (National Center of Occupational Health and Environmental Protection for Health) of the "Instituto Nacional de Salud" (National Institute of Health) from Peru.

Reply: The acknowledgment section was modified:

“None.”

Reply: The ethics statement was moved to the Methods section.

6. We note that this manuscript is a systematic review or meta-analysis; our author guidelines therefore require that you use PRISMA guidance to help improve reporting quality of this type of study. Please upload copies of the completed PRISMA checklist as Supporting Information with a file name “PRISMA checklist”.

Reply: The PRISMA checklist was add in the “Supplementary material 1. Preferred Reporting Items for Systematic reviews and Meta-Analyses extension for Scoping Reviews (PRISMA-ScR) Checklist.”

It should be noted that the check-list belongs to a scoping review.

-------

Review Comments to the Author

Reviewer #1: 

- Thanks for the opportunity to review this manuscript. The authors have undertaken a scoping review to identify that impact of nationally implemented COVID-19 mental health policies within the workplace. Please see the attached document for a few points for consideration to strengthen your review.

Reply: We reviewed and complied with the comments raised by the reviewers. How we respond to comments is described below.

- Your eligibility includes literature that “addressed national policies promoted by official governments, that were in execution or that had been executed, and that were directly or indirectly related to mental health in the workplace” - since the scope is primarily national policies, I would suggest updating your title to include this

Reply: We added local and national policies because we modified the title by:

“Policies on Mental Health in the workplace during the COVID-19 pandemic: A scoping review”

- If the goal was to identify government implemented policies, wondering if you had considered including a grey literature search within the review, as certain policy-related evaluations might have not made their way to evidence-based databases (e.g. Scopus/ Web of Science/ Embase and Pub Med)?

Reply: Our main focus is on scientific information and scientific evaluation of national policies: 

“The objective of this study is to synthesize scientific information regarding national and local policies focused on preventing or improving, directly or indirectly, mental health problems in the workplace during the COVID-19 pandemic.”

Therefore, it is not our aim to evaluate policies directly, but rather the available evidence on policies. We add in the limitations section: 

“Second, our search was conducted to identify scientific publications on the evaluation of government policies and standards for workers' mental health. Although we searched different databases, it was not within our objective to consider grey literature such as policies, standards, technical notes, or regulations from different countries. This decision may represent a partial review of the available evidence, as policy evaluations could have been performed through technical standards or internal reports; however, these documents are not peer-reviewed and are not fully reliable. Because our objective is not to evaluate policies directly, but to evaluate the available evidence on national and local policies.”

- Keeping in mind that scoping reviews aim to “map the literature on a particular topic or research area and provide an opportunity to identify key concepts; gaps in the research; and types and sources of evidence to inform practice, policymaking, and research” (Daudt et al. 2013) - since there was only a single paper included within your report, it would be worth it to do an additional synthesis of the full-text review articles that focused on national COVID-19 mental health policies (even through they did not evaluate outcomes), to truly describe the gaps in research and provide the reader with a thorough synthesis of the literature - using data from Supplementary Table 3 & 4.

Reply: We agreed with the reviewer's suggestion and reanalyzed the results of the full-text review to be able to include a larger number of papers by changing the inclusion criteria to be able to present the evidence gaps.

The criteria were modified by: “All studies that met the following characteristics were included:

- Documents that made direct or indirect reference to workplace contexts during COVID-19 pandemic

- The intervention or exposure is an implemented national and local policy and the objective was to directly or indirectly improve mental health in the work context of any occupational group. Studies on institutional policies in specific locations such as hospitals, schools or specific workplaces were excluded.

- The outcomes are any processed indicator, clinical outcome and the result of the impact of the policies, for instance, number of occupational psychology services deployed, additional amounts to health budgets, number of mental health care through tele consultations, the prevalence of mental health problems, beneficiaries’ interviews, amongst others

- The papers presented primary or secondary data (quantitative, qualitative, or mixed) focusing on mental health policy outcomes. Also, narrative reviews, systematic reviews, and meta-analyses were also included, provided they focus on national or local mental health policies in the workplace.

Papers not related to the topic explored, duplicates, documents that presented only recommendations or theoretical aspects of policies (frameworks or reference frameworks), documents that only describe the policy formulation process, and protocols were excluded.”

In the end, we were left with four studies within the scoping review that met the inclusion criteria. The results and discussion section were modified based on the new results.

- The authors outline two important limitations of it being too early to evaluate policies OR that mental health outcomes may be evaluated in larger COVID-19 strategies (instead of MH specific ones), which were also points that I had while reading through the manuscript - it could be worth to re-do the search in a year or two. 

Reply: We agree with the reviewer on the need to replicate the search in two or three years in order to allow time for new studies to be published. We add in the limitations section:

“First, because our review found limited available evidence (only one low-quality study), it is not possible to identify the policies that have the greatest impact on workers' mental health. However, this may be because it is still too early for studies to have been published on the topic, so further review on the subject is needed. The research team invites different health intelligence teams to replicate the search in two to three years, when more evidence is available, considering that during the pandemic context, the evidence becomes outdated very quickly.”

- In certain countries like Canada, where healthcare is delivered on a provincial basis, there were governmental provincial policies rather than national policies implemented, so perhaps including the justification to only focus on national policies within your limitations would be appropriate.

Reply: The inclusion criteria of the study were modified so that national and local (i.e., regional, departmental) policies can be included.

- The paper would benefit from a thorough read through for grammar: E.g. “The study protocol in PROSPERO, since this repository does not support scoping review.”

Reply: The language has been extensively revised to improve clarity and comprehension.

Reviewer #2: The article performs a systematic review on the mental health policies implemented in the workplace during the COVID-19 pandemic. Clearly, this is an important topic, useful to know to what extent companies have involved themselves in the state of their workers, as well as to examine the efficacy (or lack of it) of the employed measures.

Reply: We appreciate the reviewer's comments and will respond to them below.

- There is plenty of international evidence pointing out the mental health issues that have been increasing during the past few years (e.g. doi: 10.1186/s12992-020-00589-w). Therefore, the background must be complemented with data and references specifying the most relevant issues, clarifying the importance of the topic and its practical implications.

Reply: We agree with the reviewer, so we add a new paragraph in the background section: 

“The COVID-19 pandemic has profoundly impacted both mental health and working conditions. Various studies have reported a high prevalence of mental health disorders in the general population that have been directly and indirectly attributed to COVID-19 pandemic [9-11]. Different systematic reviews have identified an increase in the prevalence of mental health problems in many countries for the general population and workers [12, 13], which represents a problem at the public health level since a higher prevalence of mental health problems will overload the health system [14]. Likewise, a burden of mental health problems has a negative economic impact as it would reduce productivity, increase absenteeism and reduce the competitiveness of countries [15]. Therefore, it is necessary to develop strategies and policies to reduce the impact on population as a whole, especially on workers (productive companies of the country). The measures adopted by governments to mitigate the spread and transmission of the SARS-CoV-2 virus in the population, such as quarantines, blockades, social distancing, reduction of capacity, implementation of remote work, a restart of activities work in phases, among others, have drastically modified the working conditions of various sectors [16, 17]. An example of this is the health sector, which, although it never stopped its activities because it was considered as an essential sector, was seriously affected by the pandemic since a substantial increase in the workload, maintained direct exposure to COVID-19, suffered a shortage of personal protective equipment, abuse by patients, stigma, and discrimination, amongst other problems [18-20].”

- The systematic review cannot have one article only. It is true that, to the day, there are few papers analyzing the topic of policies implemented by companies. Therefore, I recommend slightly changing the eligibility criteria of the articles, in order to include at least 5 or 6 more of them in the itemization and analyses of contents. For instance, you could include studies analyzing the mental health issues derived from COVID-19 in specific populations of workers, even though they may not address the measures employed by the companies (e.g. doi: 10.7717/peerj.13050).

Reply: We agreed with the reviewer, so the inclusion criteria were changed to add national and local policies. In addition, a new review of all full-text documents was performed to corroborate that they met the inclusion criteria.

The criteria were modified by: “All studies that met the following characteristics were included:

- Documents that made direct or indirect reference to workplace contexts during COVID-19 pandemic

- The intervention or exposure is an implemented national and local policy and the objective was to directly or indirectly improve mental health in the work context of any occupational group. Studies on institutional policies in specific locations such as hospitals, schools or specific workplaces were excluded.

- The outcomes are any processed indicator, clinical outcome and the result of the impact of the policies, for instance, number of occupational psychology services deployed, additional amounts to health budgets, number of mental health care through tele consultations, the prevalence of mental health problems, beneficiaries’ interviews, amongst others

- The papers presented primary or secondary data (quantitative, qualitative, or mixed) focusing on mental health policy outcomes. Also, narrative reviews, systematic reviews, and meta-analyses were also included, provided they focus on national or local mental health policies in the workplace.

Papers not related to the topic explored, duplicates, documents that presented only recommendations or theoretical aspects of policies (frameworks or reference frameworks), documents that only describe the policy formulation process, and protocols were excluded.”

- Also, a new search must be performed in the database, since this topic is current, and articles are being constantly published. Therefore, it is probable that since October 14th some articles have been published hat adjust to the criteria established by the authors, and they will have to be included in the review.

Reply: We regret that we are unable to comply with the reviewer's suggestion because we no longer have sufficient human resources to be able to perform the update. Before the start of peer review, an update was performed at the editor's request. However, we currently do not have the resources for a new update. Therefore, we add as a limitation: 

“Third, it is possible that, at the time of publication, the findings presented may partially represent all available scientific evidence due to the speed with which new studies are published within the context of COVID-19. Thus, it warrants ongoing updates of the review by health intelligence teams throughout the pandemic.”

---

## [Decision Letter · Decision Letter 1]

18 Jul 2022

Policies on Mental Health in the workplace during the COVID-19 pandemic: A scoping review

PONE-D-21-38416R1

Dear Dr. Villarreal-Zegarra,

We’re pleased to inform you that your manuscript has been judged scientifically suitable for publication and will be formally accepted for publication once it meets all outstanding technical requirements.

Kind regards,

Sergio A. Useche, Ph.D.

Academic Editor

PLOS ONE

Additional Editor Comments (optional):

Thanks for your amendments. Your paper is now suitable for acceptance.

Reviewers' comments:

Reviewer's Responses to Questions

**Comments to the Author**

1. If the authors have adequately addressed your comments raised in a previous round of review and you feel that this manuscript is now acceptable for publication, you may indicate that here to bypass the “Comments to the Author” section, enter your conflict of interest statement in the “Confidential to Editor” section, and submit your "Accept" recommendation.

Reviewer #1: All comments have been addressed

Reviewer #2: All comments have been addressed

2. Is the manuscript technically sound, and do the data support the conclusions?

Reviewer #1: Yes

Reviewer #2: Yes

3. Has the statistical analysis been performed appropriately and rigorously? 

Reviewer #1: N/A

Reviewer #2: Yes

4. Have the authors made all data underlying the findings in their manuscript fully available?

Reviewer #1: Yes

Reviewer #2: Yes

5. Is the manuscript presented in an intelligible fashion and written in standard English?

Reviewer #1: Yes

Reviewer #2: Yes

6. Review Comments to the Author

Reviewer #1: Thank you for addressing all the comments. The updated inclusion criteria and limitations section highlight where more research is required within this field.

Reviewer #2: The authors have taken into account the suggestions I provided in my previous review, so I consider that the manuscript is suitable for publication.

7. PLOS authors have the option to publish the peer review history of their article (what does this mean?). If published, this will include your full peer review and any attached files.

Reviewer #1: No

Reviewer #2: No

---

## [Editor Report · Acceptance letter]

20 Jul 2022

PONE-D-21-38416R1 

Policies on Mental Health in the workplace during the COVID-19 pandemic: A scoping review 

Dear Dr. Villarreal-Zegarra:

I'm pleased to inform you that your manuscript has been deemed suitable for publication in PLOS ONE. Congratulations! Your manuscript is now with our production department. 

Kind regards, 

on behalf of

Dr. Sergio A. Useche 

Academic Editor

PLOS ONE